# An Optimization Ensemble for Integrated Energy System Configuration Strategy Incorporating Demand–Supply Coordination

Chenhao Sun [1] , Xiwei Jiang [1], Zhiwei Jia [1,*], Kun Yu [1], Sheng Xiang [1] and Jianhong Su [2]

1   School of Electrical and Information Engineering, Changsha University of Science and Technology, Changsha 410114, China; chenhaosun@csust.edu.cn (C.S.); jxwcsust@163.com (X.J.); kunyu0707@163.com (K.Y.); sxiang@csust.edu.cn (S.X.)
2   International College of Engineering, Changsha University of Science and Technology, Changsha 410114, China; iamjianhongsu@outlook.com
*   Correspondence: jiayege@csust.edu.cn

**Abstract:** As one representative smart energy infrastructure in smart cities, an integrated energy system (IES) consists of several types of energy sources, thus making more complicated coupling connections between the supply and demand sides than a power grid. This will impact when allocating different energy sources to ensure the appropriate energy utilization in the IES. With this motivation, an IES energy configuration optimization strategy based on a multi-model ensemble is proposed in this paper. Firstly, one coupling model is constructed to assess the underlying collaborative relationships between two sides for a renewable-energy-connected IES. Next, the independent component analysis (ICA) method is implemented for noise reduction in massive heterogeneous input databases, which can effectively improve the computing efficiency under such high-dimensional data conditions. Also, the self-adaptive quantum genetic model (SAQGM) is built for subsequent configuration optimization. Specifically, the quantum bit representation is incorporated to reduce computation complexity in multi-states scenarios, the double-chain formation of chromosomes is deployed to diminish the uncertainty when encoding, and the dynamic adaptation quantum gate is established to successively amend parameters. Finally, an empirical case study is conducted which can demonstrate the benefits of this strategy in terms of feasibility, efficiency, and economy.

**Keywords:** integrated energy system; energy configuration optimization; coupling relationships; ICA-SAQGM ensemble

## 1. Introduction

Nowadays, the integrated energy system (IES), which comes with a higher permeability of multi-type energy sources, is making notable strides in its attempt to fulfill the increasing energy consumption and eco-friendly tendency in cities. In an IES, energy utilization and dispatch strategies [1–3] can enhance the overall energy efficiency, thus achieving a high utilization rate, a high environmental protection rate, and a comprehensive energy balance between supply and demand. However, due to the complexity of coupling among the multiple heterogeneous energy sources and loads, the operation of an IES is intricate and, hence, requires further strategy optimizations.

To achieve this, mathematical programming techniques are ubiquitously utilized in energy systems. In application scenarios, the objective functions might be single or multiple, and the constraints can also be linear or nonlinear. Ergo, these tools for the IES optimization include linear programming (LP) [4], mixed-integer linear programming (MILP) [5], and mixed-integer nonlinear programming (MINLP) [6]. The mathematical programming approaches can generate precise outputs, whereas their efficiency and performance might

be impacted since the diversities and complexities grow dramatically in an IES compared with a power grid.

Hence, the machine learning (ML) methodologies are widely deployed in different fields, and their applicable range can, of course, cover the optimization of IESs. One typical group is the neural networks. An artificial neural network (ANN) was built to forecast the solar radiation and wind speed in [7]; the heating, cooling, and lighting loads in an IES were predicted via combining the ANN and the long short-term memory (LSTM) in [8]; and Chandrasekaran et al. [9] selected the ANN as the decision-making method for the operation of an IES with hybrid photovoltaic, wind power, and battery. The second is the support vector machine (SVM). The SVM was added to the firefly optimizations for solar radiation predictions in [10] and the dragonfly algorithm for wind power forecasting in [11]. Also, the least squares support vector regression (LSSVR) learning approach for hydropower consumption predictions was proposed by [12]. Another type is the random forest (RF). A tree model integration approach [13] was utilized for building energy consumption predictions, and Ahmadi et al. [14] presented a tree-based learning wind power forecasting tool. Moreover, [15] proposed an improved GM (1,1) model to improve the accuracy of load forecasting. Based on the ML techniques, one class of methods, the deep learning (DL) approaches, which consist of more hidden layers organized in deeply nested networks, are also incorporated. For instance, one DL framework for building an energy consumption forecast was presented in [16]; the short-term prediction of wind and solar was considered in the energy management model for a microgrid in [17]; Zhou et al. [18] proposed a multi-energy net load prediction method with heterogeneous prosumers; and in [19], the output of a hybrid wind–photovoltaic farm was forecasted via feature selection. These ML and DL techniques are more robust and immune to potential noise and can perform well in complex IESs. However, these methodologies generally require a larger volume of input data when theoretically analyzing, which might be difficult in some real-life scenarios. Moreover, most of them merely consider either supply or demand sides during assessments, whereas the coupling relationships between these two sides are significant in an IES as well and thus cannot be simply neglected.

On the demand side, various energy coupling devices are implemented to choose a reasonable, favorable, and efficient energy transmission approach for responding to its own energy demand. For example, in [20–22], different operational architectures of comprehensive IES demand response models were established to minimize the operation cost. The supply side, however, usually sets dissimilar energy prices to guide the optimization of energy consumption on the demand side. For instance, diverse transaction models for the IES based on energy blockchain technologies were designed in [23–25] which can not only ensure the priority of energy supply to important users but also guarantee the reliability of energy transactions. Significant progress has been brought about by the aforementioned research. Nonetheless, for the purpose of conducting more comprehensive evaluations, both the supply and demand sides ought to be coordinated. The complicated data conditions such as high-dimensional, heterogeneous, and uncertain data scenes generated from these coupling connections between two sides are indispensable and thus need to be incorporated when designing energy configuration strategies.

To this end, an IES configuration optimization strategy based on a multi-model ensemble is proposed in this paper. To start with, a coordination model between the supply and demand side in an IES is formed. Next, to cope with the random errors and biases caused by a high-dimensional heterogeneous data context, the inputs are transformed into linear combinations of multiple statistically independent components to implement the unmixing process based on the independent component analysis (ICA) model, which can achieve dimension reduction and computational efficiency strengthening. Then, the self-adaptive quantum genetic model (SAQGM) is initiated to optimize energy configurations. Firstly, the quantum bit, which refers to the basic unit of information in quantum computing, is deployed according to the quantum state characteristics, and the linear superposition of different state probabilities of each analytic solution can amend the coverage and convergence

performance, thus decreasing the number of requisite chromosomes, that is, reducing the unnecessary solutions to the problem. Secondly, a double-chain chromosome formation is designed, where each chromosome contains two concatenated genes, with each gene representing an optimal solution. Every chromosome can be delineated through a set of bimodal solutions containing biomorphic strings to handle the potential random coding and frequent decoding during the stage of quantum bits encoding. Thirdly, in the step of evolution, both the rotation angle and probability magnitude can be actively modified via the dynamic adaptation quantum gate; hence, the uncertainty can be diminished while the efficiency can be raised simultaneously. At last, the effectiveness and feasibility of the proposed strategy are verified by an empirical case study.

The contributions of this paper are summarized as follows:

1.  To handle the complicated coupling relations among supply and demand sides in an IES, this paper proposes an energy configuration optimization strategy based on a multi-model ensemble to ensure better resource allocation and arrangement, especially in more complex data distribution scenarios.
2.  To cope with the high-dimensional data conditions, the ICA approach is deployed to establish a linear transformation from high-dimensional inputs to unmixing independent components in the feature space, which can reduce the noise and thus improve computing efficiency.
3.  To solve the heterogeneous and uncertain data problems, the SAQGM methodology is designed for subsequent energy configuration optimizations, where the quantum bit representation is built to reduce computation complexity in multi-state component scenarios, the double-chain formation of chromosomes is formed to diminish the uncertainty when encoding, and the dynamic adaptation quantum gate is implemented to successively amend parameters.

The remainder of this paper is organized as follows: Section 2 demonstrates the coupling model for a target IES, Section 3 describes the proposed ICA-SAQGM ensemble, Section 4 discusses an empirical case study, and Section 5 presents conclusions of this paper.

## 2. IES Coupling Model

### 2.1. Coupling Relationships between Supply and Demand

An IES mainly includes energy input, exchange, storage, and supply [26]. The input side (supply side) is connected to various types of sources [27], and after the conversion, the energy can be supplied to the load side (demand side), such as the electric load, cold load, heat load, and natural gas load, etc. This supply network of cooling, heating, power, and natural gas are generally simultaneously connected, and the characteristics of each component are also different, making the coupling relationships between those two sides more complex. Hence, the coupling matrix can be constructed by the energy center (EC) model [28] to reflect the input–output relationships among multiple energy sources.

One representative local IES network is implemented as a test system in this paper, and the corresponding coupling relationships between the supply and demand sides are presented in Figure 1. This IES is an energy supply system with various operational features for the coordinated operation of multiple energy sources, making the network structure of this system more adaptable to a variety of energy sources and loads; thus, it can coordinate both the supply and demand sides.

As represented in Figure 1, the input energy on the supply side mainly consists of electricity, heat, and natural gas, and can be transformed into various forms of energy needed by users through energy conversion equipment such as gas turbines, electric air conditioning gas boilers, etc. This can be depicted mathematically as follows:

$$\begin{cases} P^E = \rho_1 E^E + \gamma_1^G \rho_2 E^G \\ P^H = E^H + \left(\gamma_1^G \rho_3 + \gamma_2^G \rho_4\right) E^G \end{cases} \tag{1}$$

where the $P^E$ and $P^H$ denote the supply efficiency of electrical and thermal energy in energy input; $\rho_1$, $\rho_2$, $\rho_3$, and $\rho_4$ are the efficiency of transformer input, the efficiency of electric power of gas turbine, the efficiency of heat of gas turbine, and the efficiency of gas boiler, respectively; $\gamma_1^G$ and $\gamma_2^G$ depict the distribution coefficient of the natural gas network to the gas turbine and the distribution coefficient to the gas boiler, respectively; $E^E$, $E^H$, and $E^G$ demonstrate the power network, heat network, and natural gas network resources in the energy input; and $P_W$ and $P_L$ represent the wind power and photovoltaic power connected to the integrated energy system, respectively. To ensure the linearity of this model, it is necessary to set $\gamma_1^G E^G$ and $\gamma_2^G E^G$ as the principal variables. $\gamma_1^G E^G$ indicates the amount of natural gas resources consumed by the gas turbine, and $\gamma_2^G E^G$ delineates the number of natural gas resources consumed by the gas boiler.

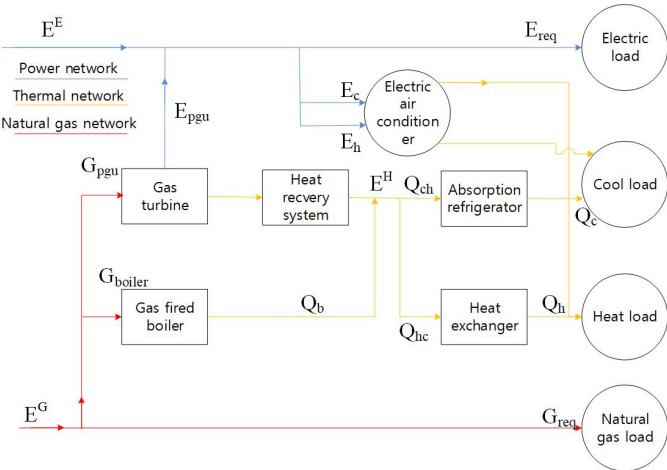

**Figure 1.** Coupling of demand and supply sides in an IES.

The EC model on the supply side can therefore be formed as follows:

$$\begin{bmatrix} P^E \\ P^H \end{bmatrix} = \begin{bmatrix} \rho_1 & 0 & \gamma_1^G \rho_2 \\ 0 & 1 & \gamma_1^G \rho_3 + \gamma_2^G \rho_4 \end{bmatrix} \begin{bmatrix} E^E \\ E^H \\ E^G \end{bmatrix} + \begin{bmatrix} P_W + P_L \\ 0 \end{bmatrix} \qquad (2)$$

On the load side, users purchase the energy on demand and convert into actual-use resources through electric air conditioners, absorption chillers, heat exchangers, and other equipment, so as to meet the requirements for various energy resources. This can be delineated mathematically as follows:

$$\begin{cases} E_{req} = \alpha^E l^E \\ Q_c = \gamma_1^E \varphi_1 l^E + \gamma_2^H \varphi_2 l^H \\ Q_h = \gamma_3^E \varphi_3 l^E + \gamma_4^H \varphi_4 l^H \end{cases} \qquad (3)$$

where $E_{req}$, $Q_c$, and $Q_h$ denote the demand for electrical energy, the demand for cold energy, and the demand for heat energy on the energy demand side, respectively; $l^E$ and $l^H$ are the overall electrical load and the overall thermal load when the energy demand is satisfied through various electrical devices, respectively; $\varphi_1$, $\varphi_2$, $\varphi_3$, and $\varphi_4$ indicate air-conditioning cooling efficiency, absorption chiller efficiency, air-conditioning heating efficiency, and heat exchanger efficiency, respectively; $\gamma^E$, $\gamma_1^E$, and $\gamma_3^E$ illustrate the distribution coefficients of direct electricity consumption equipment, air-conditioning heating state, and air-conditioning heating state in the overall electrical load, respectively; and $\gamma_2^H$, and $\gamma_4^H$ depict the distribution coefficients of absorption chillers and heat exchangers in the overall thermal load, respectively, and thus $\gamma^E + \gamma_1^E + \gamma_3^E = \gamma_2^H + \gamma_4^H = 1$.

The demand-side EC model can hence be written as follows:

$$
\begin{bmatrix} E_{req} \\ Q_c \\ Q_h \end{bmatrix} = \begin{bmatrix} \alpha^E & 0 \\ \gamma_1^E \varphi_1 & \gamma_2^H \varphi_2 \\ \gamma_3^E \varphi_3 & \gamma_4^H \varphi_4 \end{bmatrix} \begin{bmatrix} l^E \\ l^H \end{bmatrix}
\tag{4}
$$

*2.2. Coupling Relationships between Supply and Demand*

2.2.1. Economical Cost

In order to bring higher economic benefits, the operating costs should be minimized, and its objective function should be established in terms of the investment cost, the maintenance cost, the loss during the operations, and the user's comfort experience. The economical cost function can be formed as follows:

$$
min(Ec) = C_b + C_r + C_l + C_c
\tag{5}
$$

where

$$
C_b = \sum_{t=1}^{24} \left( c_t^E E_t^E + c_t^G E_t^G + c_t^H E_t^H \right)
\tag{6}
$$

In Equation (6), $C_b$ denotes the integrated system investment cost; $c_t^E$, $c_t^G$, and $c_t^H$ represent the acquisition unit price of electric network resources, natural gas network resources, and thermal network resources, respectively; and $E_t^E$, $E_t^G$, and $E_t^H$ demonstrate the amount of resources acquired by the electricity network, the amount of resources acquired by the natural gas network, and the amount of resources acquired by the thermal network, respectively.

$$
C_r = \sum_{t=1}^{24} \sum c_t P_t
\tag{7}
$$

In Equation (7), $C_r$ represents the cost to be spent by the equipment in operation, $c_t$ is the maintenance cost required for the physical equipment to output unit power per unit time period, and $P_t$ denotes the power output of the equipment per unit time period.

$$
C_l = \beta_E \left( W_E^{sup} - W_E^{req} \right) + \beta_H \left( W_H^{sup} - W_H^{req} \right) + \beta_G \left( W_G^{sup} - W_G^{req} \right)
\tag{8}
$$

In Equation (8), the $C_l$ depicts the additional cost of energy loss during operation; $\beta_E$, $\beta_H$, and $\beta_G$ denote the additional loss coefficients of energy in the operation process for electric network resources, thermal network resources, and natural gas network resources, respectively; $W_E^{sup}$, $W_H^{sup}$, and $W_G^{sup}$ delineate the supply of electric power resources, thermal power resources, and natural gas resources, respectively; and $W_E^{req}$, $W_H^{req}$, and $W_G^{req}$ represent the demand for electricity, heat, and natural gas resources, respectively.

$$
C_c = \sum_{t=1}^{24} \sum (\mu_t \cdot (L_E + L_H + L_G))
\tag{9}
$$

In Equation (9), $C_c$ is the user comfort cost, i.e., the cost expended by the user to compensate for the user's comfort not being affected when the energy demand changes; $\mu_t$ denotes the proportional coefficient of user comfort cost per unit time period; and $L_E$, $L_H$, and $L_G$ illustrate the load difference between the user's energy demand change and the initial demand.

2.2.2. Environmental Cost

When minimizing the system's operation cost, the environmental cost of energy consumption will also be taken into account. In this paper, the pollutant emissions within

system operations are selected as the measure of the environmental cost, and its objective function is built as follows:

$$min\ (En) = \sum_{t=1}^{T} \sum_{i=1}^{n} \theta_i En_i^t \tag{10}$$

where $T$ is the total annual operating time of the system, $n$ represents the number of energy types in the system, $\theta_i$ denotes the pollutant emission index of the $i$th energy source per unit time, and $En_i^t$ is the pollutant emission index of the $i$th energy source at time $t$.

### 2.2.3. Constraints

An IES involves many different types of energy, so it is important to ensure the safety and reliability of the operation of various components of the system in line with the ability to prevent impacts of extreme scenarios [29].

1.  Power balance constraint:

$$\begin{cases} P_t^{\mathrm{E}} = P_{1t}^{\mathrm{E}} + P_{2t}^{\mathrm{E}} + P_{3t} + P_{4t} \\ P_t^{\mathrm{H}} = P_{1t}^{\mathrm{H}} + P_{2t}^{\mathrm{H}} \end{cases} \tag{11}$$

where $P_t^{\mathrm{E}}$, $P_{1t}^{\mathrm{E}}$, $P_{2t}^{\mathrm{E}}$, $P_{3t}$, and $P_{4t}$ denote the output electrical power, transformer input power, gas turbine output electrical power, wind power generation power connected to the system, and photovoltaic generation power, respectively; and $P_t^{\mathrm{H}}$, $P_{1t}^{\mathrm{H}}$, and $P_{2t}^{\mathrm{H}}$ represent the output thermal power, the thermal power output from the gas turbine, and the thermal power output from the gas boiler, respectively.

2.  Equipment operation constraint:

$$\begin{cases} P_{i,min} \le P_{it} \le P_{i,max} \\ E_{i,min} \le E_{it} \le E_{i,max} \end{cases} \tag{12}$$

where $P_{it}$ denotes any device that performs energy conversion in an integrated energy system; $P_{i,min}$ and $P_{i,max}$ are the minimum and maximum values of the power output of the device, respectively; $E_{it}$ indicates the input of any network resource; and $E_{i,min}$ and $E_{i,max}$ depict the threshold value of this resource acquired by the integrated energy system.

3.  Energy price constraint:

$$c_{i,min} \le c_{it} \le c_{i,max} \tag{13}$$

where $c_{it}$ represents the acquisition price of any energy source in the system; and $c_{i,min}$ and $c_{i,max}$ indicate the lower and upper bounds of the energy price, respectively.

## 3. ICA-SAQGM Ensemble
### 3.1. ICA Model

The ICA is a method for finding the intrinsic components from a multidimensional dataset. For dataset $X$, the ICA solution can be formulated as follows:

$$X = AS \tag{14}$$

Let $X$ be a random vector and $X \in R^{m \times 1}$, and then we have the following:

$$\begin{pmatrix} x_1 \\ x_2 \\ \cdots \\ x_i \\ \cdots \\ x_m \end{pmatrix} = \begin{pmatrix} a_{11} & a_{12} & \cdots & a_{1n} \\ a_{21} & a_{22} & \cdots & a_{2n} \\ \vdots & \vdots & & \vdots \\ a_{i1} & a_{i2} & \cdots & a_{in} \\ \vdots & \vdots & & \vdots \\ a_{m1} & a_{m2} & \cdots & a_{\mathrm{mn}} \end{pmatrix} \begin{pmatrix} s_1 \\ s_2 \\ \cdots \\ s_i \\ \cdots \\ s_n \end{pmatrix} \tag{15}$$

where $s_i$ is a random variable, and both values are independent of each other. $A$ is a full rank matrix and $A \in R^{m \times n}$.

We suppose the following:

$$W = A^{-1} \tag{16}$$

thus, Equation (14) can be rewritten as follows:

$$S = \begin{pmatrix} \widetilde{w}_1 & \cdots & \widetilde{w}_j & \cdots & \widetilde{w}_n \end{pmatrix} = \begin{pmatrix} w_{11} & w_{12} & \cdots & w_{1j} & \cdots & w_{1n} \\ w_{21} & w_{22} & \cdots & w_{2j} & \cdots & w_{2n} \\ \vdots & \vdots & \ddots & \vdots & \ddots & \vdots \\ w_{i1} & w_{i2} & \cdots & w_{ij} & \cdots & w_{in} \\ \vdots & \vdots & \ddots & \vdots & \ddots & \vdots \\ w_{m1} & w_{m2} & \cdots & w_{mj} & \cdots & w_{mn} \end{pmatrix} X \tag{17}$$

where $W$ is a full rank matrix and $W \in R^{m \times n}$; and $\widetilde{w}_j$ is one included column vector.

The goal of the ICA is to use the statistically independent and non-Gaussian properties of each independent component as the objective function to find a suitable transformation matrix, $W$, thus obtaining the independent components, $S$. According to the different objective functions, the ICA can be classified into diverse types, including the large likelihood estimation and minimum mutual information method. Among them, the FastICA model deploys the negative entropy as a measure of the non-Gaussian random variables. The negative entropy will then be maximized under the constraints to solve the independent components. The benefits of this model are the fast convergence and effortless computation compared with other methods.

The flowchart of the FastICA is delineated as Figure 2:

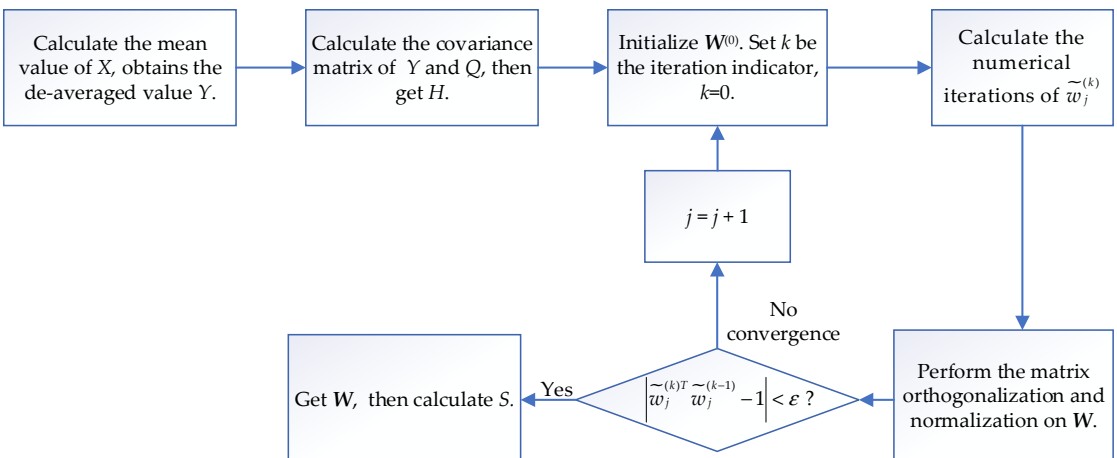

**Figure 2.** Flowchart of the FastICA.

The specific steps of the FastICA are shown below:

1. Calculate the mean value of $X$ and obtain the de-averaged value $Y$ according to the following equation:

$$Y = X - M \tag{18}$$

2. Find the covariance matrix of $Y : C = cov(Y, Y^T)$, the diagonal array of eigenvalues of the matrix $Q$, and the eigenvectors $\alpha$. Let $U = Q^{-1/2}\alpha^T$, and the processed data can be obtained via the following equation:

$$H = U \times Y \tag{19}$$

3. Initialize the random matrix, $\boldsymbol{W}^{(0)}$, and assume that the modulus of the column vector is 1. Set $k$ as the iteration indicator, such that $k = 0$.

4. Use the following equation for numerical iterations of $\widetilde{w}_j^{(k)}$:

$$\widetilde{w}_j^{(k)} = \boldsymbol{E}\left\{ H \cdot G\left( \widetilde{w}_j^{(k-1)T} \cdot H \right) \right\} - \boldsymbol{E}\left\{ H \cdot g\left( \widetilde{w}_j^{(k-1)T} \cdot H \right) \right\} \times \widetilde{w}_j^{(k-1)} \tag{20}$$

where $\boldsymbol{E}$ is the identity matrix, and $G()$ denotes the hyperbolic tangent function, that is $G(t) = tanh(\sigma t)$. Assume that the coefficient $\sigma$ is 1, and therefore $G(t) = tanh(t) = \left( e^t - e^{-t} \right) / \left( e^t + e^{-t} \right)$, and $g()$ represents the first order derivative of $G()$.

5. Perform the matrix orthogonalization and normalization on $\boldsymbol{W}$:

$$\widetilde{w}_j^{(k)} \rightarrow \sum_{i=1}^{m} \left( w_j^{(k)^r} w_i \right) w_i \tag{21}$$

$$\widetilde{w}_j^{(k)} \rightarrow \frac{w_j^{(k)}}{\left\| w_j^{(k)} \right\|} \tag{22}$$

6. When $\left| \widetilde{w}_j^{(k)T} \widetilde{w}_j^{(k-1)} - 1 \right| < \varepsilon$, then $j = j + 1$; otherwise, there is no convergence, and must return to Step 3.

7. When $j = m$, the matrix $\boldsymbol{W}$ can be generated, and then the independent components, $S$, can be figured out according to Equation (17).

From the above discussions, it is clear that the ICA is a linear transformation that can separate data samples into independent and non-Gaussian distributed linear combinations. On the whole, the ICA multiplies the independent components of the data with a decomposition matrix so that each component can reflect as much time-independent and valid information as possible without orthogonal processing. In contrast, the traditional dimensionality reduction method, the principal component analysis (PCA), performs de-correlation on the dataset based on an orthogonal mixing matrix, so that each principal component tries to cover the variance. Therefore, the ICA can be summarized as a process of unmixing linear combinations of several independent components, while the PCA is a process of information extraction, which is actually the normalization step in the ICA. Although the ICA requires a predefinition of the total number of independent components to be decomposed, i.e., it requires prior knowledge of the database, the ICA gives a better performance in regard to portraying the random statistical properties of variables and tolerating the white noise.

*3.2. SAQGM Methodology*

3.2.1. Objective Description: Quantum Bit Representation Scheme

The QGM model is generally based on the quantum bits and the concept of state superposition in quantum mechanical theory. In a system of two-state quantum, the smallest unit for information storage is called a quantum bit. A quantum bit can be the state "1" or "0" or in any superposition of these two. Thus, the state of one quantum bit can be expressed as follows:

$$\langle \Psi \rangle = \alpha \langle 0 \rangle + \beta \langle 1 \rangle \tag{23}$$

where $\alpha$ and $\beta$ are the complex coefficients representing the probability of being in the corresponding state, $|\alpha|^2$ represents the probability that the quantum bit is 0, and $|\beta|^2$ represents the probability that the quantum bit is 1.

Regularizing this complex coefficient can be achieved as follows:

$$|\alpha|^2 + |\beta|^2 = 1 \tag{24}$$

Thus, for a system with $n$ quantum bits, it will cover $2^n$ kinds of states.

In evolutionary optimization algorithms, one difference is the various representation schemes for encoding and storing. These representations can roughly be classified as the bimodal, numerical, and symbolic methods. In this paper, a representation scheme based on the concept of quantum bits is designed for the QGM.

A quantum bit is defined via a pair of complex numbers, which can be written as $(\alpha,\ \beta)^T$, where the two parameters have the same meaning as those in Equation (24).

We define a space for $m$ quantum bits as follows:

$$\begin{bmatrix} \alpha_1 & \alpha_2 & \cdots & \alpha_m \\ \beta_1 & \beta_2 & \cdots & \beta_m \end{bmatrix} \tag{25}$$

where $|\alpha_i|^2 + |\beta_i|^2 = 1, i = 1, 2, \cdots, m$.

The advantage of this representation scheme is that it can cover the superposition states other than "1" and "0". For example, the probability magnitude of a three-qubit system with three states can be written as follows:

$$\begin{bmatrix} 1/\sqrt{2} & 1/\sqrt{2} & 1/2 \\ 1/\sqrt{2} & -1/\sqrt{2} & \sqrt{3}/2 \end{bmatrix} \tag{26}$$

Then, the states of this system can be expressed as follows:

$$\frac{1}{4}(000) + \frac{\sqrt{3}}{4}(001) - \frac{1}{4}(010) - \frac{\sqrt{3}}{4}(011) + \frac{1}{4}(100) + \frac{\sqrt{3}}{4}(101) - \frac{1}{4}(110) \\ - \frac{\sqrt{3}}{4}(111) \tag{27}$$

From Equation (27), this three-qubit system contains information in eight states, namely (000), (001), (010), (011), (100), (101), (110), and (111), with probabilities of occurrence of $1/16$, $3/16$, $1/16$, $3/16/$, $1/16$, $3/16/$, $1/16$, $1/16$, and $3/16$, respectively.

Therefore, the quantum bit-based representation scheme has better performance in dealing with multi-state scenarios. As in Equation (27), one quantum chromosome, which refers to a feasible solution to the problem, can represent eight states; however, in the classical representation, at least eight chromosomes, namely (000), (001), (010), (011), (100), (101), (110), and (111), are required. In addition, the representation by quantum bits is better in terms of convergence. When coefficient $|\alpha_i|^2$ or $|\beta_i|^2$ converges to the value "1" or "0", the multi-state property of quantum bit chromosomes gradually disappears until it converges to one single state.

In summary, the population size, i.e., the number of quantum bit chromosomes, in the QGM will always remain constant, which follows the principle of quantum bit conservation in quantum computing theory. Therefore, the QGM with the quantum bit representation scheme has better diversity and convergence compared with the ordinary GM.

### 3.2.2. Quantum Bit Coding: Double-Chain Formation

In the QGM, a probabilistic analysis model is added to the standard genetic algorithm. The QGM can usually cover a quantum bit chromosome population, and the population by the $t$th generation can be expressed as follows:

$$Q(t) = \left\{ q_1^t, q_2^t, \cdots, q_l^t \right\} \tag{28}$$

where $l$ is the population size, and $q_j^t$ represents a quantum position chromosome.

In the proposed SAQGM, the double-chain coding model [30] of the $j$th chromosome is deployed, and it can be written as follows:

$$q_j^t = \begin{bmatrix} \alpha_{11}^t & \alpha_{12}^t & \cdots & \alpha_{1n}^t & \alpha_{21}^t & \cdots & \alpha_{2n}^t & \cdots & \alpha_{l1}^t & \cdots & \alpha_{mn}^t \\ \beta_{11}^t & \beta_{12}^t & \cdots & \alpha_{1n}^t & \beta_{21}^t & \cdots & \beta_{2n}^t & \cdots & \beta_{m1}^t & \cdots & \beta_{mn}^t \end{bmatrix} \tag{29}$$

where $j = 1, 2, \ldots, l$; $m$ is the number of genes in that chromosome; $n$ is the number of quantum bits, i.e., the string length of the quantum bit chromosome; each chromosome is a viable solution to the problem; and the viable solution is made up of multiple elements, which refer to the genes of the chromosome.

To initialize $Q(t)$, all $q_j^t$ in $Q(t)$ values are preset to $1/\sqrt{2}$. That is, $q_j^t\big|_{t=0}$ refers to a quantum bit chromosome that linearly superimposes all possible states with the same probability, and its state can be formed as follows:

$$\left\langle \Psi_{q_j^0} \right\rangle = \sum_{k=1}^{2^m} \frac{1}{\sqrt{2^m}} \langle S_K \rangle \tag{30}$$

where $S_K$ is the $k$th state that is represented by bimodal string $x_1, x_2, \ldots, x_m$.

For the purpose of overcoming the randomness when coding and the frequent decoding when optimizing, the quantum bit chromosome can be reformed as follows:

$$q_j^t = \begin{bmatrix} S_c(t) \\ S_s(t) \end{bmatrix} = \begin{bmatrix} \cos(\theta_{i1}) & \cos(\theta_{i2}) & \cdots & \cos(\theta_{ih}) & \cdots & \cos(\theta_{im}) \\ \sin(\theta_{i1}) & \sin(\theta_{i2}) & \cdots & \sin(\theta_{ih}) & \cdots & \sin(\theta_{im}) \end{bmatrix} \tag{31}$$

where $\theta_{ih} = 2\pi \times \tau$, and $\tau$ is a random number from 0 to 1, $h = 1, 2, \ldots, m$.

Commonly, a set of bimodal solutions $S(t)$ will be worked out through $Q(t)$, where the binary solutions are a biomorphic string with a length, $m$, and are generated via each quantum bit, $q_j^t$. Each solution can be employed to evaluate its fitness, and then the best one is selected and stored in the bistatic solution, $S(t)$. It can be noticed in Equation (29) that each chromosome $q_j^t$ contains two juxtaposed genes, and each of them represents an optimal solution set, i.e., the $S_c(t)$ and $S_s(t)$, respectively.

### 3.2.3. Evolution Strategy: Dynamic Adaptation Quantum Gate

In the iterations, the bimodal solution set $S(t)$ is calculated based on the previous time series of quantum bit chromosome populations, $Q(t-1)$. By constructing quantum gates, $U(t)$, each set of quantum bit chromosomes $Q(t)$ can be updated. $U(t)$ is built according to the bistable solutions and the optimal storage solution, which can be designed in line with dissimilar requirements. The quantum gate is chosen in this paper, and its mathematical expression is as follows:

$$\begin{bmatrix} \alpha'_{ij} \\ \beta'_{ij} \end{bmatrix} = U(\theta_{ij}) \begin{bmatrix} \alpha_{ij} \\ \beta_{ij} \end{bmatrix} = \begin{bmatrix} \cos(\theta_{ij}) & -\sin(\theta_{ij}) \\ \sin(\theta_{ij}) & \cos(\theta_{ij}) \end{bmatrix} \begin{bmatrix} \alpha_{ij} \\ \beta_{ij} \end{bmatrix} \tag{32}$$

where $\theta_{ij}$ denotes the rotation angle. Moreover, $\begin{bmatrix} \alpha'_{ij} & \beta'_{ij} \end{bmatrix}^{\mathrm{T}}$ and $\begin{bmatrix} \alpha_{ij} & \beta_{ij} \end{bmatrix}^{\mathrm{T}}$ represent the probability amplitude before and after the updating of $q_j^t$; that is, the new qubit chromosome, $Q(t)$, is obtained by rotating the qubit chromosome of the previous cycle, $Q(t-1)$.

The optimal solutions in each of them are stored in $S(t)$. We calculate their fitness and compare it with the fitness of the next set of solutions in the next cycle to ensure better solutions. If the fitness, $f(t)$, corresponding to $q_j^t$ is greater than $f(t-1)$, then the iteration proceeds in the direction that is conducive to the occurrence of $q_j^t$; otherwise, it proceeds in the opposite direction. The iteration step will repeat until the optimal solution is obtained. Ergo, the selection of $\theta_{ij}$ will significantly impact the evaluation and ought to be dynamically adjusted.

In order to achieve this in a more comprehensive manner, a statistical tool normal distribution model which can incorporate the gradient of the fitness function in the previous time series is utilized in this paper to assess the tendency of evolution. Based on the formerly calculated gradient of the fitness function, the probability density function, $\varphi(x)$, of the normal distribution model can be solved, and the variation coefficient of $\theta_{ij}$ can thus be determined via the gradient of the current evaluation function, $\nabla f(X_{ij})$. When that

gradient is low-in-value, the contemporary evolution direction is positive which requires a higher iteration speed, and otherwise lower. In that case, the self-adaptive update model for $\theta_{ij}$ can be built as follows:

$$\theta_{ij}(t+1) = \theta_{ij}(t) + sgn(x) \times \aleph \times \Delta\theta_0 \tag{33}$$

where

$\aleph = P\left(\nabla f(X) \geq \nabla f(X_{ij})\right);$

$\varphi[\nabla f(X_{ij})] = \frac{1}{\sqrt{2\pi}\sigma} e^{\frac{-(\nabla f(X_{ij})-\mu)^2}{2\sigma^2}};$

$\mu = \overline{\nabla f(X_{ij})};$

$\sigma^2 = \frac{1}{n}\sum_{i=1}^{n}\left(\nabla f(X_{ij}) - \nabla f\left(X_{(i-1)j}\right)\right)^2.$

In Equation (33), $sgn(x)$ is the update direction when $x \neq 0$, and can be positive or negative if $x = 0$, $\mu$ and $\sigma^2$ represents the average and variance gradient of $f(x)$, respectively, $\varphi[\nabla f(X_{ij})]$ is the probability density function of that gradient, $\Delta\theta_0$ denotes the initial iteration angle.

We assume that $\sigma^2$ remains constant when the input data volume grows, since the variation of it is comparably smaller than $\nabla f(X_{ij})$. Therefore, if $\nabla f(X_{ij}) \leq \mu$, the farther it locates from $\mu$, the smaller $\nabla f(X_{ij})$ will be, and the larger $\aleph$ will be, and a faster evolution can be made; otherwise, $\aleph$ will be smaller, and the evolution will also be slower. The illustration for selecting $\aleph$ is shown in Figure 3.

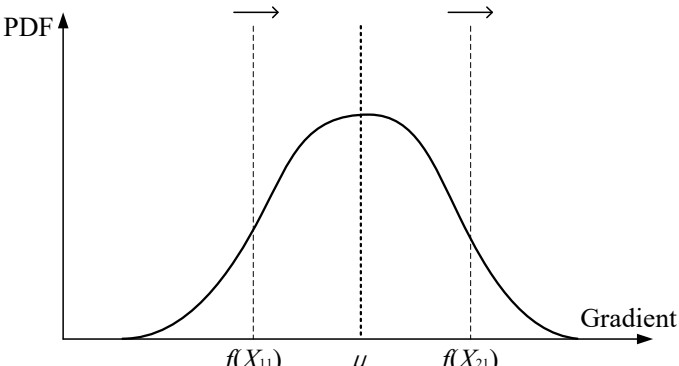

**Figure 3.** Determination of $\aleph$.

In that case, $\varphi[\nabla f(X_{ij})]$ in Equation (33) can be further simplified as follows:

$$\varphi[\nabla f(X_{ij})] = \frac{1}{\sqrt{2\pi}}\int_{-\infty}^{\nabla f(X_{ij})} e^{-\frac{g^2}{2}} dg \tag{34}$$

where $g = \frac{-(\nabla f(X_{ij})-\mu)}{\sigma}$.

Hence, $\aleph$ can be calculated as follows:

$$\aleph = 1 - \varphi[\nabla f(X_{ij})] = 1 - \frac{1}{\sqrt{2\pi}}\int_{-\infty}^{\nabla f(X_{ij})} e^{-\frac{g^2}{2}} dg \tag{35}$$

According to previous discussions, the phase of quantum bits can be adjusted via the self-adaptive, $\theta_{ij}$, and hence the double-chain coding model in Equation (31) can also be amended. The corresponding probability magnitude can be formed as follows:

$$\begin{bmatrix} \cos\theta_{ij}(t+1) \\ \sin\theta_{ij}(t+1) \end{bmatrix} = \begin{bmatrix} \cos\Delta\theta_{ij} & -\sin\Delta\theta_{ij} \\ \sin\Delta\theta_{ij} & \cos\Delta\theta_{ij} \end{bmatrix} \begin{bmatrix} \cos\theta_{ij}(t) \\ \sin\theta_{ij}(t) \end{bmatrix} = \begin{bmatrix} \cos\left[\theta_{ij}(t) + \Delta\theta_{ij}\right] \\ \sin\left[\theta_{ij}(t) + \Delta\theta_{ij}\right] \end{bmatrix} \tag{36}$$



### 3.2.4. Procedure

The main steps of the SAQGM model can be described as follows:

1. Initialize the quantum position chromosome population, $Q(t)$, with $l$ quantum position chromosomes and assume $\theta_0$ to be the initial iteration angle and the mutation probability, $P_m$, to be 0.05.
2. Probe all chromosomes, encode the quantum bits via the double-chain coding model, and calculate the fitness, as well as the gradient.
3. Evaluate the solution set, $S(t)$; solve the optimal solution; and store.
4. If the termination condition is not reached, $S(t)$ is generated via the previous time series of the chromosome population $Q(t-1)$ in each round of the iteration.
5. Adjust the rotation angle and the probability magnitude, and then update $Q(t)$ via the self-adaptive quantum gate, $U(t)$;
6. The optimal solution is stored in $S(t)$ until the termination condition is reached.

## 4. Empirical Case Study

### 4.1. Data Base

For the verification of the performance of the proposed ensemble, an IES in Eastern China is selected in this paper. The generation forecasts of wind and photovoltaic, as well as the load predictions of thermal and electric, are shown in Figure 3. The parameters of equipment and the energy prices are presented in Tables 1 and 2, respectively.

**Table 1.** Parameters of devices in the IES.

| Equipment | Efficiency | Lower Limit | Upper Limit |
|---|---|---|---|
| Transformer | Power: 0.9 | Power purchase: 0 kW | Power purchase: 4200 kW |
| Gas turbine | Power: 0.35 Heating: 0.42 | Electricity/heat production: 1000 kW/1080 kW | Electricity/heat production: 4000 kW/4200 kW |
| Gas boiler | Heating: 0.91 | Heat production capacity: 1000 kW | Heat production capacity: 6300 kW |
| Air conditioning | Cooling: 3.7 | Production capacity: 0 kW | Production capacity: 4000 kW |
| Heat exchangers | Heating: 1.2 | Production capacity: 0 kW | Production capacity: 6000 kW |
| Absorption chillers | Cooling: 1.2 | Production capacity: 0 kW | Production capacity: 4000 kW |
| External heating | Heating: 1 | Heat purchase: 0 kW | Heat purchase: 4000 kW |
| External gas supply | Air supply: 1 | Gas purchase: 0 m$^3$ | Gas purchase limit: 1350 m$^3$ |

**Table 2.** Energy prices.

| Energy Type | Parameters | Price/CNY |
|---|---|---|
| Electricity | Peak hours | 1.00 |
| | Off-peak hours | 0.60 |
| | Low hours | 0.30 |
| Heating | Upper limit | 0.75 |
| | Lower limit | 0.35 |
| Natural Gas | / | 3.33 |

### 4.2. Results Analysis

For the verification of the performance of the proposed ensemble, an IES in Eastern China is selected in this paper. The generation forecasts of wind and photovoltaic, as well as the load predictions of thermal and electric, are shown in Figure 4. The parameters of equipment and the energy prices are presented in Tables 1 and 2, respectively.

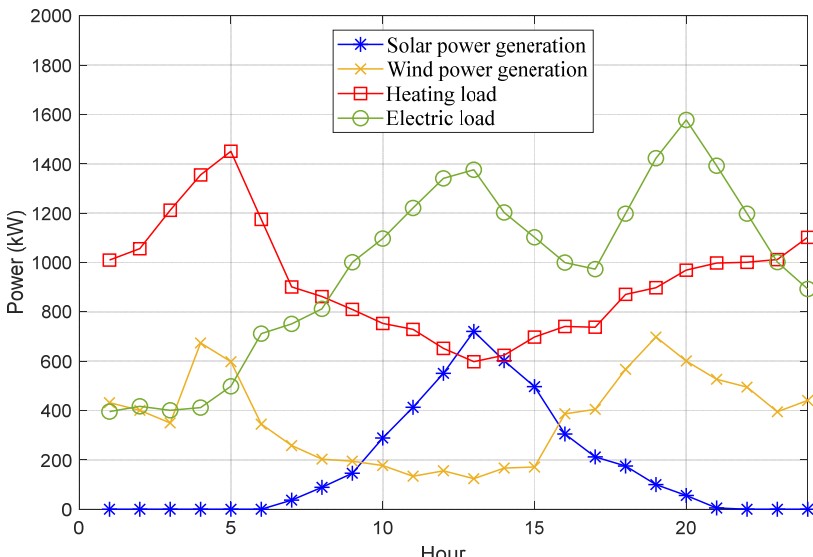

**Figure 4.** Energy forecast curves of the studied IES.

Two application scenarios are tested in this paper.

**Scenario 1:** To verify the optimization performance of the ICA-SAQGM ensemble, the particle swarm optimization (PSO) [31], the BP neural network (BPNN) [32], and the standard quantum genetic model (QGM) [33] are deployed for comparison. The maximum time of iterations of all of these algorithms is set to 300 times. The total daily cost of the studied IES is optimized, and the resulting curves are illustrated in Figure 5.

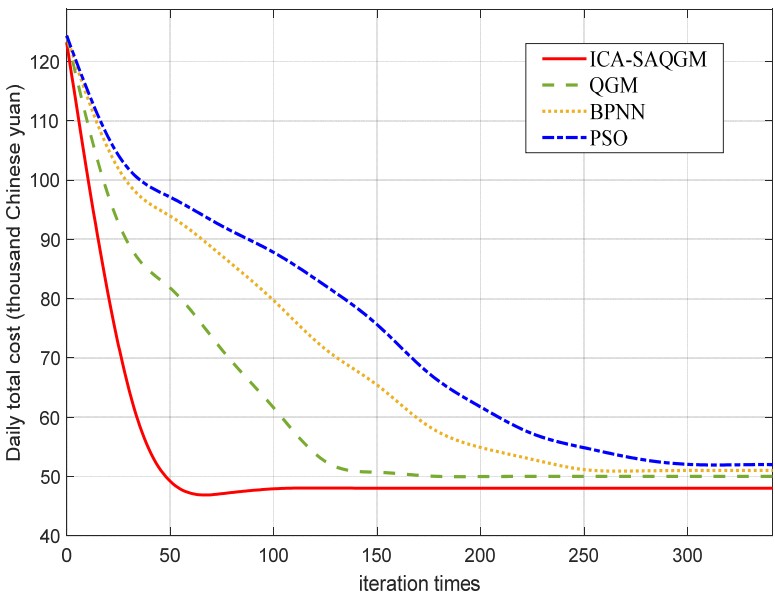

**Figure 5.** Optimization curves via four models.

In summary, evaluation indices such as the average daily cost, pollution emission, times of convergence iteration (The critical point at which the algorithm becomes stable is taken as the convergence limit), and variance are demonstrated in Table 3.

**Table 3.** Comparison of optimized consequences by different models.

| Model | Average Total Daily Cost/CNY | Average Pollution Emissions/Ton | Average Convergence Iterations | Mean Variance |
|---|---|---|---|---|
| PSO | 53,662 | 1.69 | 284 | 0.7731 |
| BPNN | 52,013 | 1.48 | 233 | 0.5215 |
| QGM | 50,978 | 1.26 | 136 | 0.2781 |
| ICA-SAQGM Model | 47,681 | 1.02 | 47 | 0.1833 |

From Figure 5 and Table 3, it can be concluded that the performance of the ICA-SAQGM is better. In terms of the average daily cost, the optimal solution of the ICA-SAQGM is 11.1%, 8.3%, and 6.5% lower than PSO, BPNN, and QGM, respectively; in terms of average pollution emissions, the ICA-SAQGM is 39.6%, 31.1%, and 19.0% better. Similarly, the proposed ensemble is finer in both the convergence speed and the variance. Above all, the ICA-SAQGM strategy can achieve a cost reduction for the IES.

**Scenario 2:** In order to assess the optimized IES considering the coupling relationship between supply and demand, three patterns are utilized: Pattern 1, standard IES operations; Pattern 2, IES operations considering the coupling relationships; and Pattern 3, optimized IES operations with the coupling relationships. Figure 5 delineates the operating curves of these three patterns.

From Figure 6a, Patterns 2 and 3 have an increase in electric loads at some specific periods, such as 1:00–7:00, since the coupling connections are incorporated into these two patterns so that the electric purchase cost can be reduced after the optimization of supply and demand. Meanwhile, in the periods with a larger electric load, such as 10:00–20:00, Pattern 3 has a better optimization effect compared with the regulation of Pattern 2, which results in a better customer satisfaction rate and better total economics. From Figure 6b, we can see that Pattern 2 has almost no regulation on the thermal load, whereas Pattern 3 makes a balance regulation of thermal loads according to the regulation of electrical loads. In Pattern 3, the combined optimization regulation of the thermal–electric loads makes it possible to cut the operating cost and to improve the overall operation of the IES without affecting users.

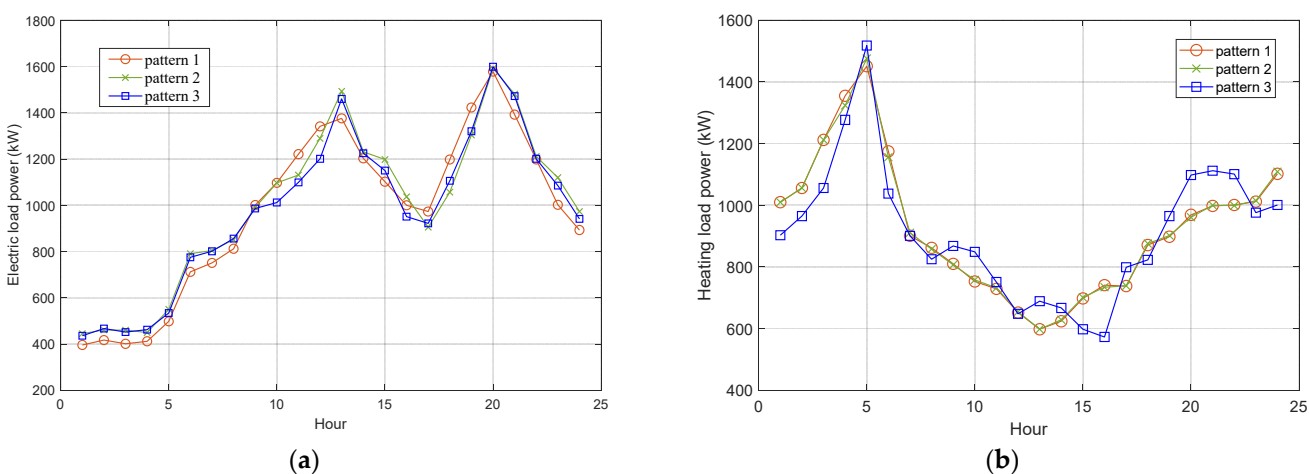

**Figure 6.** Comparison of operation curves by three patterns. (**a**) Electric load curve. (**b**) Heating load curve.

A comparison of the costs among the three patterns is provided in Table 4.

**Table 4.** Cost comparison of the three patterns.

| Pattern | Pattern 1 | Pattern 2 | Pattern 3 |
|---|---|---|---|
| Investment cost/CNY | 46,914 | 44,832 | 42,002 |
| Equipment running cost/CNY | 4522 | 3964 | 3482 |
| Loss cost/CNY | 3022 | 2552 | 1889 |
| User comfort cost/CNY | 0 | 1231 | 0 |
| Total daily cost/CNY | 54,458 | 52,579 | 47,373 |
| Pollutant emissions/ton | 1.02 | 1.02 | 1.02 |

Among these patterns, Pattern 1 does not consider the coupling relationship between supply and demand and thus assumes the energy consumption to be fixed values. In that case, the IES only needs to cover a fixed load to meet the demand; thus, the user comfort cost is 0. However, without the coupling relationships, the demand side cannot coordinate with the supply side, so the equipment operation and loss cost are both higher. Pattern 2 considers the coupling relationships, so the coordination between the demand and supply side can be achieved to decrease the equipment operation and loss cost. but will also generate extra user comfort cost. Pattern 3 can optimize Pattern 2 to more flexibly coordinate the two sides, which can reduce all the costs while not sacrificing the user's comfort. This verifies the effectiveness of the ICA-SAQGM ensemble for the IES. In addition, the pollution emissions are almost the same in different patterns, thus indicating that the coupling relationships have a small impact on this value so that it does not need further assessments.

## 5. Conclusions

In this paper, an optimization strategy based on the ICA-SAQGM ensemble is proposed for energy allocation in a renewable-energy-connected IES. The main works can be summarized as follows:

1. An IES operation model containing the coupling relationships between supply and demand sides was built which can incorporate the underlying connections between the energy sources and loads when designing strategies.
2. The ICA model was established for dimensionality reduction, which maps the high-dimensional heterogeneous inputs into the linear combinations of independent components for data unmixing. Therefore, it is used to further improve the computing efficiency.
3. The SAQGM optimization procedure was designed to enhance the comprehensiveness and efficiency, where the quantum bit representation scheme is built to decrease the computation burden in multi-state scenarios, the double-chain formation of chromosomes is deployed to ameliorate the performance when encoding, and the dynamic adaptation quantum gate was implemented to modify the parameters.
4. An empirical case study was conducted to validate the performance of the proposed method during real applications. The results prove that the configuration strategy based on this methodology can reduce investment, operation costs, and pollution emissions significantly.

**Author Contributions:** Conceptualization, methodology, software, and writing, C.S.; data curation, X.J.; supervision, Z.J.; investigation, K.Y.; validation, S.X.; reviewing, J.S. All authors have read and agreed to the published version of the manuscript.

**Funding:** This work was supported by the Natural Science Foundation of China (52207074, 52177070, and 72201041), the Natural Science Foundation of Changsha (kq2208231), and the Young Talent Project of the Provincial Education Department of Hunan (21B0334).

**Institutional Review Board Statement:** Not applicable.

**Informed Consent Statement:** Not applicable.

**Data Availability Statement:** The authors do not have the permission to share the data.

**Conflicts of Interest:** The authors declare no conflict of interest.

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
