# Peer review of "An Optimization Ensemble for Integrated Energy System Configuration Strategy Incorporating Demand–Supply Coordination"

_sustainability, doi:10.3390/su152115248_

Round 1

Reviewer 1 Report

The work presented is valuable and particularly important for practice. The optimization of integrated energy systems from a topological-functional point of view - with a reduced impact on the environment and the accumulation of minimal costs - represents a major desideratum for the development of human society. The authors demonstrate - through an empirical case study - the effectiveness of the proposed optimization model (ICA-SAQGM).

Recommendation: it would be good for the FastICA model on pages 6-7 of the paper to be reinforced by a presentation in the form of a flowchart diagram (a figure/image says more than a thousand words).

Author Response

Please see the attachment, and thanks for your work.

Reviewer 2 Report

The manuscript attempts to tackle a Herculean problem by optimizing an integrated energy system (gas, electricity, and thermal) encompassing various constraints in the planning and operation sides. Due to its complexity, the proposed approach will be significantly limited to small systems and it will be practically challenging on a large scale.  Thus, in theory, the proposed technique might be interesting to explore, as yet, its practicality is a different story. 

Nonetheless, the reviewer has a few comments:

1. The paper's contribution seems vaguely explained. With myriad studies conducted in this area, it is imperative that the authors clearly highlight the shortcomings of the presented studies and address them in their proposed work.

2. What is the rationale of eqs. (1) and (2)? Without any explanation, it is difficult for the readers to follow.

3. In the paragraph starting from line 84, there are many terms such as quantum genetic model, a quantum bit, and chromosomes, that seem to be mentioned without providing the context. Unless the readers are fully familiar with the subject at hand, it would be difficult for them to understand the flow of the manuscript presentation.

4. In line 86, what "attendant noise" is, this text referring to?

Unless the readers are deeply into this subject, it would be very hard for even technical readers can understand as so many terms are being cited without proper context and sections that are disconnected.

Author Response

(The authors gave the same response as above.)
